# Mapping sedentary behaviour (MAPS-B) in winter and spring using wearable sensors, indoor positioning systems, and diaries in older adults who are pre-frail and frail: A feasibility longitudinal study

Isabel B. Rodrigues[1,2¤]*, Suleman Tariq[3], Alexa Kouroukis[3], Rachel Swance[4], Jonathan Adachi[1], Steven Bray[5], Qiyin Fang[6], George Ioannidis[1], Dylan Kobsar[5], Alexander Rabinovich[7], Alexandra Papaioannou[1,8], Rong Zheng[9]

1 Faculty of Health Sciences, Department of Medicine, McMaster University, Hamilton, ON, Canada, 2 Department of Community Health Sciences, University of Manitoba, Max Rady College of Medicine, Winnipeg, Canada, 3 Faculty of Medicine, University of Toronto, Toronto, ON, Canada, 4 Faculty of Science, Department of Life Sciences, McMaster University, Hamilton, ON, Canada, 5 Faculty of Science, Department of Kinesiology, McMaster University, Hamilton, ON, Canada, 6 Faculty of Engineering, Department of Engineering Physics, McMaster University, Hamilton, ON, Canada, 7 Department of Surgery, Division of Orthopaedic Surgery, McMaster University, Hamilton, ON, Canada, 8 Department of Health Research Methods, McMaster University, Evidence and Impact, Hamilton, ON, Canada, 9 Faculty of Engineering, Department of Computing and Software, McMaster University, Hamilton, ON, Canada

¤ Current address: Department of Medicine, University of McMaster, Hamilton, Ontario, Canada
* rodrigib@mcmaster.ca

## Abstract

Older adults who are frail are likely to be sedentary. Prior interventions to reduce sedentary time in older adults have not been effective as there is little research about the context of sedentary behaviour (posture, location, purpose, social environment). Moreover, there is limited evidence on feasible measures to assess context of sedentary behaviour in older adults. The aim of our study was to determine the feasibility of measuring context of sedentary behaviour in older adults with pre-frailty or frailty using a combination of objective and self-report measures. We defined "*feasibility process*" using recruitment (20 participants within two-months), retention (85%), and refusal (20%) rates and "*feasibility resource*" if the measures capture context and can be linked (e.g., sitting-kitchen-eating-alone) and are all participants willing to use the measures. Context was assessed using a wearable sensor to assess posture, a smart home monitoring system for location, and an electronic or hard-copy diary for purpose and social context over three days in winter and spring. We approached 80 potential individuals, and 58 expressed interest; of the 58 individuals, 37 did not enroll due to lack of interest or medical mistrust (64% refusal). We recruited 21 older adults (72±7.3 years, 13 females, 13 frail) within two months and experienced two dropouts due to medical mistrust or worsening health (90% retention). The wearable sensor, indoor positioning system, and electronic diary accurately captured one domain of context, but the hard copy was often not completed with enough detail, so it was challenging to link it to the other devices. Although not all participants were willing to use the wearable sensor, indoor

**Data Availability Statement:** All relevant data are within the manuscript and its Supporting Information files.

**Funding:** IBR received the New Investigator Fund from the Hamilton Health Science to support the MAPS-B project [NIF-22541], as well as the 2022 MIRA-AGE-WELL Award and the 2022 CIHR Fellowship Award. This study was also funded by the Natural Sciences and Engineering Research Council of Canada's (NSERC) Collaborative Research and Training Experience (CREATE) program in the form of support for the Smart Mobility of the Aging Populations project which was received by RZ.

**Competing interests:** NO authors have competing interests.

positioning system, or electronic diary, we were able to triage the measures of those who did. The use of wearable sensors and electronic diaries may be a feasible method to assess context of sedentary behaviour, but more research is needed with device-based measures in diverse groups.

## Introduction

Older adults who are frail are more likely to be sedentary [1,2]. Frailty is a multidimensional syndrome characterized by a decline in function across multiple physiological systems including the cardiovascular, musculoskeletal, neurological, and immunological systems [3,4]. Sedentary behaviour is defined as any activity during awake hours in a seating, reclining, or laying posture that uses low energy expenditure (i.e., $\leq 1.5$ metabolic equivalent of task [MET]) [5]. Sedentary behaviour is not merely the absence of moderate or vigorous physical activity but also a reduction in sit-to-stand transitions, stand time, and light physical activity [6]. Most older adults who are frail spend 60% of their awake time in a seated or laid position [6]. Prolonged periods of sedentary time can lead to muscle and bone unloading and are associated with declines in mobility and quality of life, and increased risk of falls, fractures, and death [1,6–9]. In addition, prolonged screen-based sedentary activities are associated with both depressive and metabolic syndromes [2]. The deleterious health effects of sedentary behaviour are different to those of physical inactivity and are partially independent of an individual's physical activity levels [6]. Even older adults who meet the recommended aerobic exercise guidelines of moderate to vigorous physical activity might experience adverse effects of sedentary behaviour [6]. Thus, interventions to reduce periods of prolonged sedentary behaviour are necessary, especially among older adults who are living with frailty.

Prolonged sedentary behaviours are a recognised risk factor for many medical disorders, which makes it an urgent objective for preventative health interventions. To evaluate the effectiveness of such interventions, measures that are responsive to change are required [10]. Although accelerometer-derived assessments indicate that older adults have the highest levels of sedentary time [11], these objective measures do not provide contextual information to identify interventions or public health messages to reduce sedentary time [10,12]. Inclinometers are the most sensitive and valid measure of total sedentary time, but the limitation of such devices is its inability to accurately assess specific modalities of sedentary behaviour [13] Moreover, device-based measures have a high cost-to-utility ratio, which often limits their use in research [14]. A recent meta-analysis reported that current tools for assessing context of sedentary behaviour or total sedentary time either over-report or under-report the amount of time adults and older adults spend sitting [12]. For example, single item self-report questionnaires typically underestimate sedentary time when compared to device-based measures (accelerometers and inclinometers) [12]. On the other hand, multi-item questionnaires, ecological momentary assessments, and diaries with a short recall period are more accurate at measuring sedentary time; however, there is also a high degree of variability between and within those tools [12]. Currently, there is no gold standard to assess the context of sedentary behaviour, especially in older adults.

Almost all studies in older adults have assessed total sedentary time, which does not provide enough information to understand the context of sedentary behaviours [2,8]. The main reason to understand context is because not all sedentary behaviours need to be modified as some cognitively engaging sedentary behaviours (e.g., reading, socializing) appear to benefit health,

while time spent in more passive activities may be detrimental. A sedentary behaviour research priorities international consensus statement suggests researchers should explore objective and self-report methods to assess context of sedentary behaviour among older adults [8]. We used the Sedentary Behaviours International Taxonomy to guide our definition of context of sedentary behaviour [15]. Context was defined as the purpose of the sedentary behaviours, the location where the behaviours occur, posture of the behaviours (e.g., lying, sitting), social context (e.g., alone or with others), and time of day the behaviours occur. To map the context of sedentary behaviour we used objective (i.e., accelerometer and home monitoring system with an indoor positioning system), and self-report (i.e., diary) measures; we chose three measures as one measure alone does not provide enough information about context. Our study is unique because it uses a combination of measures to assess context of sedentary behaviour; however, the feasibility of these combined measures in older adults is unknown. The primary purpose of our study was to determine the feasibility of using three measures to assess the context of sedentary behaviour in older adults who are pre-frail and frail. Our secondary objectives to quantify the context of sedentary behaviours [16] and to understand perspectives of sedentary behaviour [17] are reported elsewhere.

## Materials and methods

### Study design

We conducted a mixed-methods longitudinal study with older adults who are pre-frail and frail. We followed the STROBE 2007 guidelines for reporting of observational studies (S1 Table) [18]. Ethics approval was obtained from the Hamilton Integrated Research Ethics Board. We registered our study with clinicaltrials.gov (NCT05661058) on December 22nd, 2022. We assessed feasibility over three days (one weekend and two weekdays) in the winter and spring as sedentary behaviour may differ by the season.

### Setting

We recruited participants from physicians' offices, the local newspaper, and a local radio station. We also posted advertisements on social media using Facebook and Twitter. To ensure diversity in our recruitment process we partnered with CityHousing Hamilton Corporation, an organization that provides subsidized housing to low-income older adults, many of whom are of visible minorities, immigrants, and have visible disabilities (i.e., use a walker or cane). The results of our recruitment and retention strategy of diverse (members of racial and ethnic minorities, diverse genders, low socioeconomic status) are described elsewhere [19]. We recruited participants between January to February 2023. We obtained written informed consent from each participant prior to enrolling them in the study. Participants attended two study visits (once in the winter and another in the spring) in a private room at St. Peter's Hospital, which is part of the Hamilton Health Sciences. We provided free transportation for participants with limited mobility or free parking at the hospital. Participants were grouped into four cohorts of five participants. During the first week, we met with five participants at St. Peter's where they completed a series of questionnaires and physical performance measures. We provided each participant with a wearable sensor, explained how to set up and calibrate the indoor positioning system, and complete the electronic or hard copy diary. During the second week, we collected the devices and diaries and transferred the data to a McMaster University cloud, and cleaned and charged the devices. We repeated the process with each cohort and the entire process was repeated in the spring. Participants with limited mobility or transportation were provided with pre-paid boxes to return study items. At the end of the study,

participants received remuneration as a gift card to an easily accessible location on the bus route with versatile buying options (e.g., groceries, clothing, furniture, cleaning supplies).

## Participants

We included participants if they: 1) spoke English or attended with a translator or caregiver; 2) were ≥60 years and older; and 3) had a Morley Frail Scale score ≥3 (i.e., a score of 0 is robust, a score 1 or 2, pre-frail, and a score of 3 to 5, frail) [20]. We excluded individuals who: 1) used a wheelchair for at least 55% of the awake day due to medical conditions; 2) were not independently mobile (i.e., require assistance from another individual to ambulate); and 3) had travel plans or other commitments that required missing >30% of the rollout period. We sought to enroll both males and females as we anticipated that gender may influence sedentary behaviour through socially constructed norms and roles and can be affected by differential access to resources, opportunities, and power.

## Measures and data sources

To map the context of sedentary behaviour we used objective (wearable sensor and indoor positioning system), and self-report (daily diary) measures. Participants were equipped with the wearable sensor and indoor positioning system, and completed a diary of daily activities over three days (one weekend and two weekdays) in the winter (February 1, 2023 to March 21st, 2023) and spring (April 10th, 2023 to May 27th, 2023). The three measures were linked using date and time (e.g., sitting-living room-watching TV-alone weekend, Winter 3:30 pm to 5:15 pm).

**Wearable sensor.**   We used the activPAL4™ to collect data on posture. The activPAL4™ is a valid tool to use among older adults that generates totals for the time spent lying, sitting, standing, and stepping every second of the day [21]. The wearable sensor was secured to participant's right upper thigh, midway between the iliac crest and the upper line of the patella, using a waterproof 3M Tegaderm Transparent bandage. Participants were asked to continue their normal daily activities as the wearable sensor would not interfere with their daily lives. Data was collected on the device's hard drive and exported manually to a secure McMaster University cloud. We considered a valid wear day if the participant wore the monitor for the full 24-hour of inclinometer wear time for at least three days that included two weekdays and one weekend.

**Indoor positioning system.**   We used a custom-designed and developed indoor positioning system to obtain room level positioning information. The system was designed and validated to be used by older adults in their own homes without the need for a floor plan and only minimal initial setup and calibration; the system can also be used in homes with multiple stories with multiple residents [22]. The indoor positioning system consists of a smartwatch, a few beacons, and a data hub. The participants wore a commercially available, off-the-shelf smartwatch with customized software. The smartwatches were waterproof and could be used in the shower and pool. The location of the smartwatches is tracked by ambient (nonwearable) beacons plugged in regular wall outlets of different functional rooms of the participant's homes; we defined functional rooms as areas that participants used at least 25% of the day (e.g., kitchen, bedroom, living room). The system detected location and tracked the room-to-room movements of the participants at seconds intervals [22]. The data was collected wirelessly by a data hub and stored on a secured McMaster cloud data server.

**Diary.**   Each participant was asked to complete a diary of 24-hour daily activities using an electronic diary (Activities Collected over Time over 24-hours (ACT24)) or a hard copy version that asked participants to describe their activity, who they did the activity with, and the

date and time. ACT24 was developed by the National Cancer Institute for research purposes [23,24]. ACT24 is an internet-based previous-day recall designed to estimate total time (hours/day) spent sleeping in bed, in sedentary behaviours during awake hours, and in physical activity [23,24]. ACT24 also provides estimates of energy expenditure associated with each behaviour (MET-hours/day) [23,24]. We provided all participants with several sheets of the hard copy diary and participants who used the electronic diary inputted their activities the next day into ACT24. We sent daily email reminders to participants to complete their electronic diary.

**Health outcomes.**   We collected baseline data on falls in the last 6-weeks, cognition score, frailty status, activities of daily living, health-related quality of life, depression, and anxiety in the winter and spring. Fall history was assessed by asking the following question: "*we would like to know about any falls you have had in the last 6-weeks. Have you had any fall including a slip or trip in which you lost your balance and part or all of your body landed on the floor or ground or lower level?*" [25]. Cognitive status was assessed using the Montreal Cognitive Assessment (MoCA); we administered version 8.2 English in winter and MoCA Basic in spring [26]. MoCA scores were adjusted for age and education level. Frailty scores were measured using the Fit-Frailty Assessment & Management Application (pre-frail scores 0.18 to 0.24 and frail >0.24) [27], activities of daily living with the Nottingham Extended Activities of Daily Living Scale [28], and health-related quality of life using the EuroQol 5-Dimension 5-Level (EQ-5D-5L) questionnaire [29]. We assessed depression scores using the Geriatric Depression Scale [30] and anxiety using the Geriatric Anxiety Scale (GAS-10) [31]. Demographic characteristics were collected using PROGRESS (Place of residence, Race/ethnicity, Occupation, Gender and sex, Religion, Education, Socioeconomic status, and Social capital) [32].

## Sample size

As the primary outcome is feasibility, we selected a sample size of 20, which was considered large enough to understand the practicability of using this novel approach to mapping sedentary behaviour. Sample sizes between 12 to 24 are considered reasonable for feasibility and pilot studies [33,34].

## Outcomes

Our primary outcome was feasibility, which was defined using "feasibility process" and "feasibility resource" [35]. Feasibility process included recruitment, retention, and refusal rates, while feasibility resource was determined using the following questions: 1) can each measure capture its intended domain of context (e.g., does the diary capture purpose and social context); 2) can data be triaged by date and time; and 3) are all participants willing to use or complete the measures. Our criteria for success for feasibility process were to recruit 20 participants within two-months with 85% retention and 20% refusal rates. Our recruitment criterion is based on previous frailty research in which 1-in-5 individuals who are approached in clinic are successfully recruited [25,36]. We anticipated that the physicians could approach 10 potential participants per week for 8 weeks (80 total participants). Our criteria for retention and refusal rates were based on a frailty systematic review where retention rates range from 70% to 90% and refusal rates from 10% to 20% [37]. Our criteria for success for feasibility resources were determined if each measure could capture a domain of context, where if "yes" than feasibility is achieved, while if "no" or "sometimes", feasibility is not achieved [35]. The same dichotomous methods were applied if the measures could be triaged using date and time (yes or no/sometimes), and if participants were willing to use activPAL4$^{TM}$, the indoor positioning system, and complete the ACT24 (yes or no/sometimes for each measure). We also conducted exit interviews with each participant to ask about experiences using each measure.

## Statistical analysis

If demographic data, feasibility process, and feasibility resources were normally distributed, we reported the results using means and standard deviations or as a count and percentage; if data was not normally distributed, we reported it as a median and interquartile range (IQR). The Shapiro-Wilk Test was used to determine normality. Descriptive analyses were performed using Microsoft Excel (version 16.71). Each exit interview was audio-recorded, transcribed verbatim, and analysed using content analysis [38]. Missing values were reported as missing. Individuals who were loss to follow-up were included in the analysis if their data were available. Adverse events were reported using narrative description.

## Results

### Feasibility process

We approached 80 individuals, and 58 expressed interest in the study (Fig 1). Of the 58 individuals, 37 declined (64% refusal rate) to enroll citing lack of interest because they initially

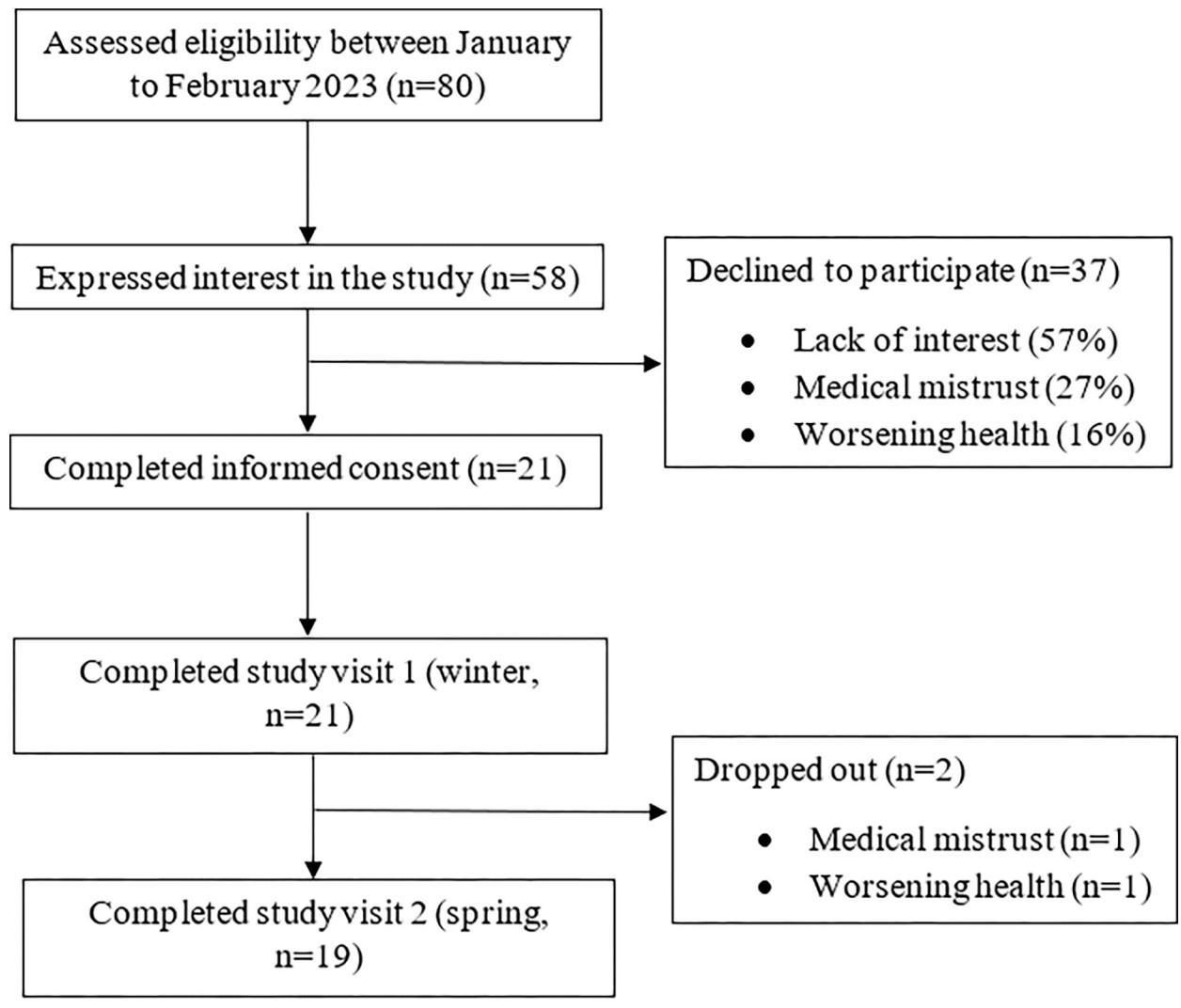

**Fig 1. Flow diagram of recruitment and retention process.**

thought the study was an exercise trial or they changed their mind (57%), medical mistrust (27%), or worsening medical health (16%). Twenty-eight of the 37 individuals who declined to participate identified as female, 1 as transgender male, and all 37 individuals had a Morley frail score $\geq$ 3. We enrolled 21 participants within two months. About 71% of participants were recruited from a physician's office, 19% from advertisements posted in CityHousing Hamilton, and the other 10% from community advertisements posted on social media or the radio. A day after the initial study visit, one participant withdrew citing medical mistrust in the study and another participant withdrew after completing the winter period citing worsening medical health (90% retention rate). Both individuals who were lost to follow up were over the age of 75 years and categorized as frail. Five participants required transportation and three utilized the pre-paid box option.

## Participant characteristics

Baseline characteristics are presented in Table 1. Participants who were frail had poorer scores on the Geriatric Depression Scale, Geriatric Anxiety Scale, gait speed, and 5x sit-to-stand compared to individuals who were pre-frail. There were no differences between the frail and pre-frail group on the MoCA, grip strength, EQ-5D-5L, and the Nottingham Activities of Daily Living. We also found no differences between physical performance measures and health outcomes between the winter and spring. One participant did not complete the MoCA in winter because they forgot their reading glasses, and another did not complete the Geriatric Depression Scale in spring for personal reasons.

## Feasibility resource

We found that each measure captured its domain of context except for the hard copy diary. The hard copy diary was not completed with sufficient information about the activity or time of day. We were able to link data from activPAL4™, the indoor positioning system, and ACT24, but not with the hard copy diary. All participants were willing to complete the hard copy diary while almost all participants were willing to use the wearable sensor. Only some participants opted to use the indoor positioning system and complete ACT24.

**Wearable sensor.** Twenty of the 21 participants felt comfortable using the activPAL4™ device to assess posture. One participant initially agreed to wear the device, and then removed it immediately after the study visit citing medical mistrust. All 20 participants found the device "comfortable to wear" and that they "did not really notice it". Participants wore the devices continuously for three or four days to capture two weekdays and one weekend. Some participants were initially concerned the 3M Tegaderm would cause skin irritation, but we experienced no adverse events. From a research perspective, the devices were easy to set up, extract data, and charge. The median wear days was 4 days ($Q_1$ = 3, $Q_3$ = 4, n = 19) in the winter and spring. During the winter session, the wearable sensor was not set up properly for one participant, and so data on posture was not collected and considered invalid.

**Indoor positioning system.** Six of the 21 participants were willing to use the indoor positioning system. The other 15 participants were not prepared to try the system as they anticipated challenges in the set up, which directly requires connection to their home WiFi router. These participants expressed concerns including not familiar with a home WiFi router, or difficulty accessing the router because it was in a hard-to-reach area. We also learned that four of the 15 participants used a cellular network (Long-Term Evolution system), which was not compatible with our version of the hub design. Overall, the six participants found the indoor positioning system easy to use but provided suggestions to improve the functional design. All six participants reported that the watch was "bulky" and "uncomfortable". The watch required

**Table 1. Demographic and other health characteristics of participants at baseline (Winter) (n = 21).**

| | |
|---|---|
| Mean age (SD), years | 73 ± 7.3 |
| Mean height (SD), cm | 166.7 ± 11.2 |
| Mean weight (median, Q1-Q3), kg | 72.6, 63.12–82.63 |
| BMI (median, Q1-Q3) | 24.02, 22.95–28.92 |
| Female sex, n (%) | 13 (62%) |
| Ethnicity, n (%) | |
| Caucasian | 18 (85%) |
| South Asian | 2 (10%) |
| East Asian | 1 (5%) |
| Highest Level of Education, n (%) | |
| Grade school | 5 (24%) |
| High school | 6 (28%) |
| Higher education (college or university) | 10 (48%) |
| Employment, n (%) | |
| Retired | 19 (10%) |
| Medical leave | 1 (32%) |
| Full-time (40 hours/week) | 1 (10%) |
| Annual income, 2023 CAD | |
| <20,000 | 2 (10%) |
| 20,001 to 40,000 | 7 (32%) |
| 40,001 to 60,000 | 2 (10%) |
| >60,000 | 10 (48%) |
| Place of Residence, n (%) | |
| In the community alone | 8 (38%) |
| In the community with others | 12 (57%) |
| Retirement home, alone | 1 (5%) |
| Visit from friends and family, n (%) | |
| Daily | 8 (38%) |
| Weekly | 8 (38%) |
| Monthly | 5 (24%) |
| Medical history, n (%) | |
| Cancer | 6 (29%) |
| Cardiovascular | 4 (19%) |
| Hearing impairment | 8 (38%) |
| Joint disease | 11 (52%) |
| Musculoskeletal condition | 9 (42%) |
| Respiratory | 5 (24%) |
| Frail Score mean (SD) | 0.35 ± 0.08 |
| Frail, n (%) | 13 (62%), 0.4 ± 0.1 |
| Pre-Frail, n (%) | 8 (38%), 0.2 ± 0.03 |
| EQ-5D-5L Utility Score, (median, Q1-Q3) | 0.78, 0.70–0.90 |
| EQ-5D-5L Visual Analogue Scale, (median, Q1-Q3) | 75.00, 57.50–87.50 |
| GAS-10, (median, Q1-Q3) | 5.50, 1.00–10.5 |
| Geriatric Depression Scale, (median, Q1-Q3) | 2.00, 0.50–6.00 |
| MoCA | 21.33 ± 4.37, n = 20 |
| Nottingham Activity of Daily Living, (median, Q1-Q3) | 17.50, 14.25–21.00, n = 20 |
| Falls in the last 6 weeks, n (%) | 6 (29%) |
| 1 fall | 3 (50%) |
| > 1 fall | 3 (50%) |
| Sedentary behaviour over 24-hours: | |
| Laying (hours), (median, Q1-Q3) | 8.50, 7.85–9.33, n = 19 |
| Sitting (hours) | 9.80, 8.20–11.65, n = 19 |
| Light physical activity over 24-hours: | |
| Standing (hours) | 4.10, 237–5.92, n = 19 |
| Walking (hours) | 1.20, 0.90–1.68, n = 19 |
| Step count | 4588, 3600–7443, n = 19 |

daily charging and so we ask participants to charge the watch overnight in the room where they slept. Most participants reported few challenges with setting up the beacons or the hub but found the black box design could be improved to be a brighter colour to make the devices less intimidating. Participants were unsure if the beacons and hub were working as there was no indicator light. From a research perspective, linking the data was a little challenging as several participants forgot to calibrate the devices; however, we were able to link it to the other measures. In addition, we know of one participant who switched one beacon to another room mid-way through data collection.

**Diary.** We collected 115 days of diary data of the 120 days (6 days x 20 participants). Eighteen participants submitted six days of diary data (three days in winter and three days in spring). One participant only submitted three days in the winter and then dropped out due to medical reasons, while the second participant only submitted four days total as they forgot to complete two days in the spring. Four participants had more than 5 hours of missing data per day; these four participants submitted a hard copy diary. Nine participants reported their daily activities using the ACT24 while the other 12 used a hard copy diary. Initially four participants agreed to use ACT24, but due to challenges in using the software, they decided to complete the hard copy instead. Challenges in using ACT24 included it being "difficult" and "complicated" at first because of the "five-min interval reporting". Some participants found it challenging to navigate because there were so many options for activities; however, after some practice the majority of participants found ACT24 "fairly easy". Participants who used the hard copy diary found it easy to complete; four participants had a caregiver complete their hard copy diary. From a research perspective, the hard copy diaries were not a feasible method to collect data as they were not completed with enough detail to extract time, purpose, and social context. The advantage of ACT24 is participants cannot submit an incomplete entry, which encouraged participants to provide enough details about their daily activities. Adherence to the diaries was good with all 20 participants completing either the electronic or hard copy diary probably because they received daily email reminders.

## Adverse events

We experienced three adverse events among two participants, which were not related to the study. One participant fell twice due to improper footwear or stepping out of the shower onto a damp floor. Another participant with type II diabetes skipped breakfast and felt unwell during the study visit; after consuming orange juice, the person returned to baseline.

## Discussion

We conducted a study in older adults who were pre-frail and frail to understand the feasibility to measure the context of sedentary behaviour in the winter and spring. Context was assessed using a wearable sensor (activPAL4$^{TM}$, posture), a McMaster engineering home monitoring system (indoor positioning system, functional location within the home), and a diary (ACT24 or hard copy diary, purpose and social environment). We met our criteria for recruitment and retention but experienced high refusal rates mainly due to lack of interest or medical mistrust. We found that each measure captured context of sedentary behaviour except for the hard copy diary. Since the hard copy diary was not completed with sufficient detail, we found it challenging to link it to the other two measures. We were able to link data from activPAL4$^{TM}$, the indoor positioning system, and ACT24. All participants were willing to complete the hard copy diary while almost all participants were willing to use the wearable sensor. Only some participants opted to use the indoor positioning system and complete ACT24. The use of

wearable sensors and electronic diaries may be a feasible method to assess context of sedentary behaviour, but more research is needed with device-based measures in diverse groups.

It is unclear what are the best measures to assess context of sedentary behaviour, especially in older adults. Objective measures, such as inclinometers or thigh-worn accelerometers, offer the highest validity for measuring sedentary time, although these measures are not able to identify the specific types of sedentary behaviours [39]. On the other hand, subjective measures, particularly diaries, are useful for recording the type and amount of time spent engaging in different sedentary behaviours, but their validity in gauging total sedentary time is low [39]. Using both objective and subjective approaches together can yield a more comprehensive measure of sedentary behaviour than one measure alone as they capture several domains of sedentary behaviour [39]. Although there is ample amount of data on devices to capture sedentary behaviour or time, there is little information about the feasibility of using a combination of measures in older adults [40]. Our results present interesting findings that suggest inclinometers and electronic diaries may be a feasible method to assess context of sedentary behaviour; however, the results are not generalizable to diverse older populations (e.g., lower socioeconomic status, visible minorities, lower education levels). There is evidence from qualitative research that suggest older adults living with chronic conditions perceive wearable activity trackers to be "useful" and "acceptable" [41–43]. But our study found that diverse older adults did not feel comfortable using any wearable device including the indoor positioning system or the wearable sensor. In fact, the most common reason for not joining the study was the fear of being tracked. The homogenous demographic characteristics of the participants in our study should be considered when interpreting the results. The majority of participants that partook in the study felt comfortable using the wearable sensor, but several participants were not willing to configure the smart home system because they were intimidated with the system and set-up process. Black boxes were used to set up the indoor positioning Bluetooth® system and several individuals found these boxes to be unsettling as they believed the boxes contained cameras. Another challenge in setting up the system was connecting the modem to the internet box. We found participants preferred to use the hard copy diary over the electronic diary, but the hard copy diary was not completed with enough detail making it challenging to link the time to the inclinometer. Those that chose to complete the electronic diary found it time consuming as they could not submit their diary if there were missing times in the day. Despite the challenge, participants found the electronic diary easy to use after enough practice. Our results suggest the combination of wearable sensors and electronic diaries may be a feasible method to capture context of sedentary behaviour; however, more research is needed to understand other methods to assess context of sedentary behaviour in diverse populations.

Smart home monitoring systems may be a potential device to assess context of sedentary behaviour. Artificial intelligence, machine learning, and fuzzy logic can be automatically rendered within smart home monitoring systems and be used to identify activities that older adults engage in (e.g., watching TV in the living room). One study developed a robot-integrated smart home (RiSH) for older adults, which used a sensor network to monitor body activities. The RiSH was able to recognize 37 distinct individual activities through sound actions with 88% accuracy and identify falling sounds with 80% accuracy [44]. Moreover, smart home systems could also be used to target and decrease certain sedentary behaviours. Rudzicz and colleagues developed a mobile robot to assist older adults with Alzheimer's disease with their activities of daily living [45]; such systems could be used to promote safe mobility among older adults who are frail. There may be an advantage to using smart home monitoring systems that utilize artificial intelligence, machine learning, and fuzzy logic as it decreases burden on participants to constantly monitor their day-to-day activities in minute-by-minute intervals. However, introducing such technologies also requires educating certain

groups that may be mistrustful of the devices. Educational outreach programs and involving diverse groups as patient partners during the co-design process should be conducted in parallel with pilot studies of smart home monitoring systems.

To date there are no set standards for the use of wearable devices with respect to wear time (minimal or maximum) or position of the device [46]. Some studies suggest that hip-worn wearable devices assess 24-hour movement more precisely than wrist-worn devices [12], whereas other investigators report reasonable precision with wrist-worn devices [12,14]. The methods researchers use to assess sedentary behaviour with wearable devices are dependent on the study aim, the design of the wearable device, the activity that is aimed to be captured, as well as the acceptability of the study population [46]. To date, most studies have used a single, objective measure to assess total sedentary time in older adults with wear time ranging from two to seven days [40,47–49]. There are few papers that used a combination of inclinometers and other measures to assess context of sedentary behaviour [50,51], which makes it challenging when selecting a wear time that accurately captured sedentary behaviour. A 2015 cohort study by Leask and colleagues claims to be the first study to explore the context of sedentary behaviour in older adults (46). The study employed a combination of a timelapse camera (Vicon Revue[TM], formerly known as SenseCam) and an inclinometer (activPAL[TM]) [50]. The average wear time for the devices was 1.5 days, with a median wear time of one day [50]. After discussions with the research team and patient partners, we decided to collect six days total with three days in the winter and three days in the spring. It was recommended by our patient partners that data collection for each season be limited to three days as to decrease the burden on participants when completing the daily diaries. It was discussed that as most individuals who are frail also have diminished cognitive impairments, the burden to accurately complete the diaries would be high. In addition, evidence from Marshall et al [52] has previously reported there are no significant differences between weekday or weekday and weekend sedentary behaviour in older adults, so we expected six days of activity would be sufficient.

## Strengths and limitations

Our study had several strengths. We recruited a diverse group of older adults who were mainly frail and had cognitive impairments with diverse demographic characteristics including individuals who only completed grade school or high school. We also used a unique combination of objective and subjective measures to assess context of sedentary behaviour. While our study conformed to the highest standards, our study is not without its limitations. The disadvantage of using only one wearable sensor can result in device failure or corrupt data; we experienced one instance where the data was not captured during the winter period. Although we attempted to recruit diverse individuals (e.g., ethnic minorities, individuals of different genders), we experienced barriers including medical mistrust. Thus, the generalizability of the results may not be feasible in other groups. As this was a feasibility study, we only collected data over three days (two weekdays and one weekend) in the winter and spring, which may not be representative of the season or other time periods (i.e., summer and fall). Moreover, it is possible that three days per season is not enough to capture the diversity of day-to-day activities of older adults who are frail and so we need more data on wear time methods and how seasonality may influence day-to-day activities.

## Conclusion

We met our criteria for recruitment and retention but experienced high refusal rates. We recruited 21 older adults who were pre-frail or frail within two months and experienced two dropouts due to medical mistrust or worsening health. We experienced high refusal rates as

several participants who initially agreed to participate decided not to enroll. The wearable sensor, indoor positioning system, and ACT24 accurately captured one domain of context but participants experienced challenges completing the hard copy diary. The hard copy was not completed with enough details making it difficult to link it to the other devices. We also found some participants were not willing to utilize the wearable sensor, indoor positioning system, and electronic diary. However, we were able to triage the measures of participants who utilized the wearable sensor, indoor positioning system, and ACT24. Nevertheless, there is some merit to using a combination of assessment methods (e.g., wearable sensor and electronic diary) to capture the context of sedentary behaviour. Future studies will need to determine the most feasible and valid methods to assess the context of sedentary behaviour, especially in diverse older adults.

## Supporting information

**S1 Table. 2007 Strobe checklist for cohort studies.**
(PDF)

**S1 File. TREND checklist.**
(PDF)

**S2 File. Protocol MAPS-B.**
(PDF)

**S3 File. De-identified feasibility data.**
(XLSX)

## Acknowledgments

The authors would like to thank our patient partners, Margaret Denton and Anne Pizzacalla from the Hamilton Council on Aging, and Priscilla Ching from Osteoporosis Canada for their input during the study. We would also like to thank our participants for helping collect this data.

## Other information

This review was registered on clinicaltrials.gov (NCT05661058).

## Author Contributions

**Conceptualization:** Isabel B. Rodrigues.

**Data curation:** Isabel B. Rodrigues, Suleman Tariq, Alexa Kouroukis, Rachel Swance.

**Formal analysis:** Isabel B. Rodrigues.

**Funding acquisition:** Isabel B. Rodrigues, Qiyin Fang.

**Investigation:** Isabel B. Rodrigues, Steven Bray, Qiyin Fang, George Ioannidis, Dylan Kobsar, Alexander Rabinovich, Alexandra Papaioannou, Rong Zheng.

**Methodology:** Isabel B. Rodrigues, Jonathan Adachi, Qiyin Fang, Dylan Kobsar, Rong Zheng.

**Project administration:** Isabel B. Rodrigues.

**Resources:** Isabel B. Rodrigues, Jonathan Adachi, Qiyin Fang, Alexander Rabinovich.

**Software:** Isabel B. Rodrigues, Qiyin Fang.

**Supervision:** Isabel B. Rodrigues.

**Writing – original draft:** Isabel B. Rodrigues.

**Writing – review & editing:** Isabel B. Rodrigues, Rachel Swance, Jonathan Adachi, Steven Bray, Qiyin Fang, George Ioannidis, Dylan Kobsar, Alexander Rabinovich, Alexandra Papaioannou, Rong Zheng.

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
