## [Decision Letter · Decision Letter 0]

28 Nov 2023

PONE-D-23-23330Mapping the context of sedentary behaviour (MAPS-B) using wearable sensors, indoor positioning systems, and diaries in older adults who are pre-frail and frail: A feasibility studyPLOS ONE

Dear Dr. Rodrigues,

Thank you for submitting your manuscript to PLOS ONE. After careful consideration, we feel that it has merit but does not fully meet PLOS ONE’s publication criteria as it currently stands. Therefore, we invite you to submit a revised version of the manuscript that addresses the points raised during the review process.

We look forward to receiving your revised manuscript.

Kind regards,

Dimitrios Sokratis Komaris, Ph.D

Academic Editor

PLOS ONE

“The authors would like to thank the following funding agents for their support with the project including McMaster Institute for Research on Aging (MIRA), AGE-WELL, the Hamilton Health Sciences New Investigator Fund, and the Canadian Institutes of Health Research. The corresponding author, who is also the principal investigator, is funded by the 2022 MIRA-AGE-WELL Award and the 2022 CIHR Fellowship Award. We would also like to thank our patient partners, Margaret Denton and Anne Pizzacalla from the Hamilton Council on Aging, and Priscilla Ching from Osteoporosis Canada for their input during the study.“

“IBR received the New Investigator Fund from the Hamilton Health Science (NIF-22541) to support the MAPS-B project. IBR also received funding from the 2022 MIRA-AGE-WELL Award and the 2022 CIHR Fellowship Award. The funders did not play any role in the study design, data collection or analysis , decision to publish, or preparation of the manuscript.“

5. We note that the original protocol that you have uploaded as a Supporting Information file contains an institutional logo. As this logo is likely copyrighted, we ask that you please remove it from this file and upload an updated version upon resubmission.

Reviewers' comments:

Reviewer's Responses to Questions

**Comments to the Author**

1. Is the manuscript technically sound, and do the data support the conclusions?

Reviewer #1: Yes

Reviewer #2: Yes

Reviewer #3: Yes

2. Has the statistical analysis been performed appropriately and rigorously? 

Reviewer #1: Yes

Reviewer #2: Yes

Reviewer #3: No

3. Have the authors made all data underlying the findings in their manuscript fully available?

Reviewer #1: No

Reviewer #2: Yes

Reviewer #3: No

4. Is the manuscript presented in an intelligible fashion and written in standard English?

Reviewer #1: Yes

Reviewer #2: Yes

Reviewer #3: Yes

5. Review Comments to the Author

Reviewer #1: This is an interesting study, looking at the feasibility of mapping sedentary lives.

Some minor comments:

1. The sample size is 21, and means(SD) have been presented but data is likely to be skewed. Was this checked? In that stats analysis section, suggested to include median(IQR).

2. Table 1: Suggested to add BMI, an indicator of BMI status, will inform of the population included in analyses.

3.Almost half of the population have joint disease/Musculoskeletal condition, are these not likely to confound with the sedentary behaviour?

Reviewer #2: 1. In the abstract and the conclusion , the authors said "Future studies will need ...", which was not the answer of the object of the study (the aim of study was determine the feasible). The authors want to say that this method was not feasible?

2. In the discussion part, the 2nd and 3rd paragraphs seems to be redundant, including many review of the papers. The authors should mention mainly discussion of results of the present study.

Reviewer #3: The presented work considers the important issue of monitoring sedentary lifestyles in older population, while being a feasibility study, it maps the road for future work. The manual exhibits commendable strengths in its approach to sample diversity as well as considering a range of metrics to assess an optimal approach to evaluation of sedentary behaviours in older adults in future full scale studies. It is well written and structured.

There are some concerns that, in my opinion, should have been addressed in the methodology section as well as considered as the limitations of the current study.

The consideration of trial duration is neglected in the manuscript, and while the authors do note a high variability issue for wearable sensor measurements, they fail to elucidate why a specific observation length (2 weekdays, 1 weekend) was selected in the methodology. The limitations highlighted concentrate on seasonal variations but omit discussions on the minimum required length of trial for meaningful results. This oversight disregards the potential bias stemming from the high variability as well as potential impact of being observed on the study subjects’ behaviour.

According to (Aadland E, Ylvisåker E. Reliability of Objectively Measured Sedentary Time and Physical Activity in Adults. PLoS One. 2015) achieving an ICC of 0.80 often requires more than one week of measurement when employing wearables in sedentary behaviour assessment, highlighting the necessity for researchers to be cognizant of intra-individual variability in accelerometer measurements. Similarly, in older adults (Gardiner et all, Measuring older 454 adults’ sedentary time: Reliability, validity, and responsiveness. Med Sci Sports Exerc. 455 2011) acceptable level of test-retest reliability (ρ =0.52 [0.27 to 0.70]) and validity (ρ=0.30 [0.02-0.54]) was reported for six day tirlas.

The use of shorter trials can lead to an overly optimistic retention prediction. Considering that actual trials can potentially be of longer duration, this oversight impacts the assessment of feasibility. The discussion should extend to the optimal method selection for longer trial periods or ways to improve reliability while staying within the 3-day range, incorporating considerations of participant retention.

Additionally, have authors considered that values might not be normally distributed and reporting only means would yield an incomplete picture, considering no other data is publicly available without request?

I believe the abovementioned points are appropriate when considering the feasibility of the study and will enhance the overall quality and usefulness of the work if addressed.

6. PLOS authors have the option to publish the peer review history of their article (what does this mean?). If published, this will include your full peer review and any attached files.

Reviewer #1: No

Reviewer #2: No

Reviewer #3: No

---

## [Author Response · Author response to Decision Letter 0]

30 Jan 2024

Thank you for reviewing and providing feedback on our manuscript "Mapping of sedentary behaviour (MAPS-B) in winter and spring using wearable sensors, indoor positioning systems, and diaries in older adults who are pre-frail and frail: A feasibility longitudinal study". Revisions are noted as tracked changes within the manuscript. The corresponding page and line numbers are associated with the track changed manuscript.

Reviewers' comments:

Reviewer #1: This is an interesting study, looking at the feasibility of mapping sedentary lives.

Some minor comments:

1. The sample size is 21, and means (SD) have been presented but data is likely to be skewed. Was this checked? In that stats analysis section, suggested to include median (IQR).

To address your comment, we conducted the Shapiro-Wilk Test and the following outcomes were not normally distributed: weight, BMI, health-related quality of life measured using the EQ-5D-5L, Visual Analogue Scale (EQ-5D-5L), anxiety score (GAS-10), depression score (GDS), and activities of daily living (Nottingham Activities of Daily Living). We included the following changes to the methods: “If demographic data, feasibility process, and feasibility resources were normally distributed, we reported the results using means and standard deviations or as a count and percentage; if data was not normally distributed, we reported it as a median and interquartile range (IQR). The Shapiro-Wilk Test was used to determine normality.” (page 8, lines 226-229). We updated Table 1 (page 10, line 258 to include median and IQR (Q1 – Q3). 

2. Table 1: Suggested to add BMI, an indicator of BMI status, will inform of the population included in analyses.

We included BMI in Table 1 (page 10, line 259). BMI was not normally distributed and was reported as a median and IQR.

3.Almost half of the population have joint disease/Musculoskeletal condition, are these not likely to confound with the sedentary behaviour?

This is a really good point; however, in our study we found that despite medical conditions such as having pain from osteoarthritis, older adults were willing to engage in activities they enjoyed despite the pain (see Theme 3 at the following link: https://www.researchsquare.com/article/rs-3315592/v1). Our results align with other findings such as the Tam-Seto 2016 and McEwan 2017 papers on perceptions of sedentary behaviour in healthy community dwelling older adults. For example, McEwan and colleagues found that physical health is only a perceived disadvantage of engaging in sedentary activities but did not actually affect their behaviour (https://pubmed.ncbi.nlm.nih.gov/26874187/). 

Reviewer #2: 

1. In the abstract and the conclusion , the authors said "Future studies will need ...", which was not the answer of the object of the study (the aim of study was determine the feasible). The authors want to say that this method was not feasible?

The original interpretation of the sentence was that our results suggest the combination of wearable sensors and electronic diaries may be a feasible method to capture context of sedentary behaviour; however, more research is needed to understand other methods to assess context of sedentary behaviour in underrepresented populations. Our study attempted to include individuals from diverse populations, and we found that individuals with certain characteristics were not comfortable using the wearable devices or indoor positioning system (see citation 17 in the manuscript). To clarify this point, we have removed the original sentence and included the following: “The use of wearable sensors and electronic diaries may be a feasible method to assess context of sedentary behaviour, but more research is needed with device-based measures in diverse groups.” (Page 2, lines 27-49). 

2. In the discussion part, the 2nd and 3rd paragraphs seems to be redundant, including many review of the papers. The authors should mention mainly discussion of results of the present study.

After rereading the discussion, we agree with reviewer 2 that the current discussion is redundant. We have deleted paragraphs 2 and 3 from the discussion and instead included a discussion of the feasibility of using objective and subjective measures to assess sedentary behaviour, other potential assessments to measure sedentary behaviour in diverse populations, and standards of wearable devices with respect to wear time protocols. We included the new discussion on page 13 to 16, lines 333 to 403. 

Reviewer #3: The presented work considers the important issue of monitoring sedentary lifestyles in older population, while being a feasibility study, it maps the road for future work. The manual exhibits commendable strengths in its approach to sample diversity as well as considering a range of metrics to assess an optimal approach to evaluation of sedentary behaviours in older adults in future full scale studies. It is well written and structured.

1. There are some concerns that, in my opinion, should have been addressed in the methodology section as well as considered as the limitations of the current study. The consideration of trial duration is neglected in the manuscript, and while the authors do note a high variability issue for wearable sensor measurements, they fail to elucidate why a specific observation length (2 weekdays, 1 weekend) was selected in the methodology. The limitations highlighted concentrate on seasonal variations but omit discussions on the minimum required length of trial for meaningful results. This oversight disregards the potential bias stemming from the high variability as well as potential impact of being observed on the study subjects’ behaviour. According to (Aadland E, Ylvisåker E. Reliability of Objectively Measured Sedentary Time and Physical Activity in Adults. PLoS One. 2015) achieving an ICC of 0.80 often requires more than one week of measurement when employing wearables in sedentary behaviour assessment, highlighting the necessity for researchers to be cognizant of intra-individual variability in accelerometer measurements. Similarly, in older adults (Gardiner et all, Measuring older 454 adults’ sedentary time: Reliability, validity, and responsiveness. Med Sci Sports Exerc. 455 2011) acceptable level of test-retest reliability (ρ =0.52 [0.27 to 0.70]) and validity (ρ=0.30 [0.02-0.54]) was reported for six day trials.

Thank you for brining up this really important point. During discussions with the research team and patient partners, it was brought up that 7-days is the typical length of wear time to collect data with the inclinometers; however, there are also several studies in older adults that used < 7 days of wear time. To inform our decision, we used a systematic review by Skender et al 2016, which found that wear time in older adults ranged from 2 to 7 days. Our team discussed that the methods researchers use to measure sedentary behaviour are dependent on the study aim, design of the device, activity that is to be captured, and acceptability within the study population. The patient partners on our team suggested that we limit data collection to six days total (three days in the winter and three in the spring) to decrease the burden of completing the daily diary as most participants who are frail also live with a cognitive impairment. We also discussed that day-to-day activities of older adults would not be substantially different according to a 2015 General Social Survey on Time Use by the Government of Canada. As most older adults are retired, three days per season was considered reasonable compared to adult wear time where day to day activities may be substantially different. To address your point and the potential limitation in our study we included the following paragraph in the discussion: “To date there are no set standards for the use of wearable devices with respect to wear time (minimal or maximum) or position of the device (44). Some studies suggest that hip-worn wearable devices assess 24-hour movement more precisely than wrist-worn devices (12), whereas other investigators report reasonable precision with wrist-worn devices (12,14). The methods researchers use to assess sedentary behaviour with wearable devices are dependent on the study aim, the design of the wearable device, the activity that is aimed to be captured, as well as the acceptability of the study population (44). To date, most studies have used a single, objective measure to assess total sedentary time in older adults with wear time ranging from two to seven days (38,45–47). There are few papers that used a combination of inclinometers and other measures to assess context of sedentary behaviour (48,49), which makes it challenging when selecting a wear time that accurately captured sedentary behaviour. A 2015 cohort study by Leask and colleagues claims to be the first study to explore the context of sedentary behaviour in older adults (46). The study employed a combination of a timelapse camera (Vicon RevueTM, formerly known as SenseCam) and an inclinometer (activPALTM) (48). The average wear time for the devices was 1.5 days, with a median wear time of one day (48). After discussions with the research team and patient partners, we decided to collect six days total with three days in the winter and three days in the spring. It was recommended by our patient partners that data collection for each season be limited to three days as to decrease the burden on participants when completing the daily diaries. It was discussed that as most individuals who are frail also have diminished cognitive impairments, the burden to accurately complete the diaries would be high. In addition, evidence from Marshall et al (50) has previously reported there are no significant differences between weekday or weekday and weekend sedentary behaviour in older adults, so we expected six days of activity would be sufficient.” (page 15 to 16, lines 387 to 409). 

2. The use of shorter trials can lead to an overly optimistic retention prediction. Considering that actual trials can potentially be of longer duration, this oversight impacts the assessment of feasibility. The discussion should extend to the optimal method selection for longer trial periods or ways to improve reliability while staying within the 3-day range, incorporating considerations of participant retention.

We also added the following sentence in the Strengths and Limitations section: “Moreover, it is possible that three days per season is not enough to capture the diversity of day-to-day activities of older adults who are frail and so we need more data on wear time methods and how seasonality may influence day-to-day activities.” (pages 16 and 17, lines 423 to 426).

3. Additionally, have authors considered that values might not be normally distributed and reporting only means would yield an incomplete picture, considering no other data is publicly available without request?

Review 1 brought up the same point. This was our response: “The original interpretation of the sentence was that our results suggest the combination of wearable sensors and electronic diaries may be a feasible method to capture context of sedentary behaviour; however, more research is needed to understand other methods to assess context of sedentary behaviour in underrepresented populations. Our study attempted to include individuals from diverse populations, and we found that individuals with certain characteristics were not comfortable using the wearable devices or indoor positioning system (see citation 17 in the manuscript). To clarify this point, we have removed the original sentence and included the following: “The use of wearable sensors and electronic diaries may be a feasible method to assess context of sedentary behaviour, but more research is needed with device-based measures in diverse groups.” (Page 2, lines 27-49). 

4. I believe the abovementioned points are appropriate when considering the feasibility of the study and will enhance the overall quality and usefulness of the work if addressed.

Thank you for your suggestions. We hope our changes are to your satisfaction.

---

## [Decision Letter · Decision Letter 1]

27 Feb 2024

Mapping sedentary behaviour (MAPS-B) in winter and spring using wearable sensors, indoor positioning systems, and diaries in older adults who are pre-frail and frail: A feasibility longitudinal study

PONE-D-23-23330R1

Dear Dr. Rodrigues,

We’re pleased to inform you that your manuscript has been judged scientifically suitable for publication and will be formally accepted for publication once it meets all outstanding technical requirements.

Kind regards,

Dimitrios Sokratis Komaris, Ph.D

Academic Editor

PLOS ONE

Additional Editor Comments (optional):

Reviewers' comments:

Reviewer's Responses to Questions

**Comments to the Author**

1. If the authors have adequately addressed your comments raised in a previous round of review and you feel that this manuscript is now acceptable for publication, you may indicate that here to bypass the “Comments to the Author” section, enter your conflict of interest statement in the “Confidential to Editor” section, and submit your "Accept" recommendation.

Reviewer #1: All comments have been addressed

Reviewer #2: (No Response)

Reviewer #3: All comments have been addressed

2. Is the manuscript technically sound, and do the data support the conclusions?

Reviewer #1: Yes

Reviewer #2: Yes

Reviewer #3: Yes

3. Has the statistical analysis been performed appropriately and rigorously? 

Reviewer #1: Yes

Reviewer #2: N/A

Reviewer #3: Yes

4. Have the authors made all data underlying the findings in their manuscript fully available?

Reviewer #1: Yes

Reviewer #2: Yes

Reviewer #3: Yes

5. Is the manuscript presented in an intelligible fashion and written in standard English?

Reviewer #1: Yes

Reviewer #2: Yes

Reviewer #3: Yes

6. Review Comments to the Author

Reviewer #1: (No Response)

Reviewer #2: The manuscript has been revised accordance with reviewers' comments. If possible, results of feasibility could be summarized as Tables.

Reviewer #3: The comments were adequately addressed. Thank you for the detailed reasoning behind the choice of time interval. The overall manuscript is, in my opinion, improved.

Please address minor typos/inconsistencies in the final version of the S5 file (supplementary materials), e.g., Participant 9 spring weight and height are mixed up.

7. PLOS authors have the option to publish the peer review history of their article (what does this mean?). If published, this will include your full peer review and any attached files.

Reviewer #1: No

Reviewer #2: No

Reviewer #3: No

---

## [Editor Report · Acceptance letter]

29 Apr 2024

PONE-D-23-23330R1 

PLOS ONE

Dear Dr. Rodrigues, 

I'm pleased to inform you that your manuscript has been deemed suitable for publication in PLOS ONE. Congratulations! Your manuscript is now being handed over to our production team.

Kind regards, 

on behalf of

Dr. Dimitrios Sokratis Komaris 

Academic Editor

PLOS ONE